# Taylor State Merging at SSX: Experiment and Simulation

**Michael Brown** \*,†**, Kaitlin Gelber** †  **and Matiwos Mebratu** †

Department of Physics and Astronomy, Swarthmore College, Swarthmore, PA 19081-1397, USA;
kgelber1@swarthmore.edu (K.G.); mmebrat1@swarthmore.edu (M.M.)
\* Correspondence: doc@swarthmore.edu
† These authors contributed equally to this work.

**Abstract:** We describe experiments and simulations of dynamical merging with two Taylor state plasmas in a Swarthmore Spheromak Experiment (SSX) device. Taylor states are formed by magnetized plasma guns at opposite ends of the device. We performed experiments with Taylor states of both senses of magnetic helicity (right-handed twist or left-handed twist). We present results of both counter-helicity merging (one side left-handed, the other right-handed) and co-helicity merging (both sides left-handed). Experiments show significant ion heating, consistent with magnetic reconnection. We suggest that the merged, warm state could be a suitable target for future magneto-inertial fusion experiments. Magnetohydrodynamic simulations of these experiments reveal the structure of the final relaxed, merged state.

**Keywords:** magnetic reconnection; magnetohydrodynamics; magneto-inertial fusion

## 1. Introduction

A Taylor state [1,2] is the term for a relaxed magnetic plasma structure with a large aspect ratio ($\ell/R \gg 1$). It is the minimum energy state of magnetized plasma. Taylor states in the Swarthmore Spheromak Experiment (SSX) begin as axisymmetric spheromaks (with poloidal and toroidal fields) but quickly evolve via instabilities to a twisted, non-axisymmetric structure bounded by a cylindrical copper liner inside the vacuum chamber. The key benefit of Taylor states is that they are stable; they have already undergone the instabilities that plague other configurations. Large aspect ratio Taylor states were first studied in the SSX device at Swarthmore [3,4]. We have since accelerated and compressed Taylor state plasmas while measuring local proton temperature $T_i$, plasma density $n_e$, and magnetic field **B**. A particular equation of state (EOS) was identified in these experiments [5–7]. Magnetothermodynamics is the study of compression and expansion of magnetized plasma with an eye towards identifying equations of state. The physics of magnetothermodynamics as well as the turbulent plasma properties of Taylor states were first elucidated at the SSX magnetohydrodynamics (MHD) wind tunnel at Swarthmore College [8,9].

Magneto-inertial fusion (MIF) experiments involve the heating and compression of magnetized plasmas [10,11]. The compression is often performed mechanically by either physically imploding a liner [12], or collapsing a liquid metal wall [13]. Recently, small scale (4.65 mm diameter, 10 mm long), compression experiments were carried out at the magnetized liner inertial fusion (MagLIF) experiment at Sandia. In these experiments, laser produced plasmas are heated in 100 ns from 100 eV to 4 keV, and magnetic fields are amplified from 10 T to 1000 T [14,15]. In the MagLIF experiments, compression is performed with a 20 $MA$ liner implosion, with converging velocities up to 70 km/s with a few 100 μg of fuel. Up to $2 \times 10^{12}$ fusion neutrons have been measured per pulse. Mechanical compression studies have been performed at SSX [5–7], but in recent experiments and simulations, we studied the dynamical

merging of two Taylor state plasmas, seeking to form a hot, stable, stagnant configuration that could be a suitable MIF target. A stable, warm magnetized target plasma is needed for future MIF fusion experiments. In our case, converging velocities are about 60 km/s, also with about 100 µg of fuel.

In Section 2, we review the SSX experiment and diagnostics in the recent merging configuration. In Section 3, we discuss the experiment and results. Finally, in Section 4, we discuss simulation results using the Dedalus framework.

## 2. SSX Taylor State Merging Configuration

### 2.1. Experimental Setup

The Swarthmore Spheromak Experiment (SSX) is currently configured to study dynamical merging of two large aspect ratio Taylor state plasmas. In the present configuration, the SSX device features a $\ell \cong$ 1.0 m long, high vacuum chamber in which we generate high density $n_e \geq 10^{15}$ cm$^{-3}$, hot $T_i \geq 20$ eV, highly magnetized $B \leq 0.5$ T hydrogen plasmas (See Figure 1). The protons are strongly magnetized: $\rho_i \approx 1$ mm, which is small compared to the dimensions of the machine ($\ell/R = 0.86$ m/0.08 m $\cong 10$). Taylor states with either sign of helicity can be formed on either side of the device. Dynamical merging ensues at the midplane with careful timing of the firing of the plasma guns. If reconnection occurs at the midplane, then we might expect the newly merged structure to retain the original twist of the two Taylor state plasmas in the case of co-helicity merging. We might expect the twist to "unravel" in the case of counter-helicity merging (i.e., the final state should have no net twist).

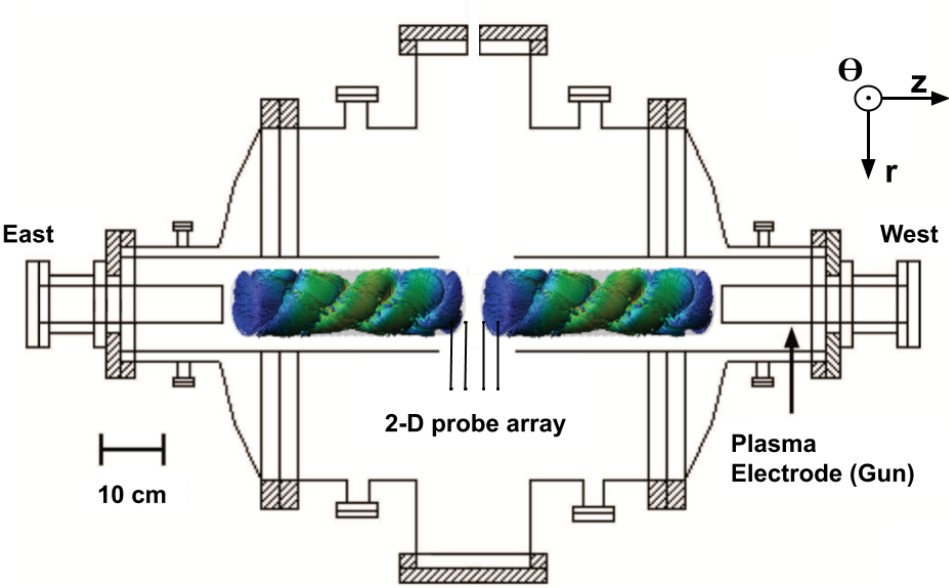

**Figure 1.** Schematic of the Swarthmore Spheromak Experiment (SSX) device in the merging configuration. Taylor state plasmas are launched by magnetized coaxial plasma guns. The merging region is outfitted with a 2D $\dot{B}$ probe array to measure magnetic field structure. In addition, co-located ion Doppler spectroscopy and HeNe laser interferometry chords are used for measuring the ion temperature and plasma density, respectively, at the midplane.

The initial conditions for all experiments considered here were the same, with the exception of the initial helicities. Plasma guns were prepared with 1.0 mWb of flux (about 1 T in a 1.5 in diameter rod of Permandur). Guns were pressurized with a rapid puff of $H_2$ (99.9999% pure) for a total of $\leq 10^{20}$ protons. The timing delay was carefully scanned between the guns to determine the optimum merging at the midplane. The hydrogen was ionized with 4 kV on each gun with 1 mF capacitors on either side (8 kJ of stored energy on either side). We performed several 30-shot campaigns with either

counter-helicity merging (one gun generating a right-handed twist object, and the other left-handed), or co-helicity merging (both guns generating left-handed objects).

Once SSX plasmas are formed in the coaxial plasma guns, they are accelerated by large $\mathbf{J} \times \mathbf{B}$ forces in the gun ($10^4$ N) with discharge currents up to 100 kA, acting on small masses (100 µg, or about $10^{20}$ protons). Ejection velocities are in the range $\cong 20 - 40$ km/s. Plasmas flow into a highly evacuated ($10^{-8}$ torr), field-free, cylindrical target volume. For these studies, we opened a 0.18 m gap at the midplane where we focus our diagnostic attention (magnetics, $T_i$, $n_e$, see below). The volume near the guns is bounded by a highly conducting thick copper shell ($r = 0.08$ m, thickness 6.3 mm) which serves as a conserver of magnetic flux. The inner plasma-facing surface is operated at 120 °C. We found that a hot plasma-facing surface tends to reduce accumulation of cold gas on the walls.

The plasma ejected out of the gun is unstable to a tilt instability and turbulently relaxes to a twisted magnetic structure [1–4]. We use the initial relaxation phase to study MHD turbulence, though this work focuses on the merging of two fully formed plasma objects. The parameters match those of earlier studies in which the magnetic structure of the object was confirmed by detailed measurements [3,4]. The plasma evolves to an equilibrium that is well described by a non-axisymmetric, force-free state (Taylor state) despite finite plasma pressure ($\beta \approx 40\%$) and large flow speeds ($M = 1$).

## 2.2. Diagnostics

We used a $4 \times 4$ magnetic probe array at the midplane to measure magnetic field structure [16,17]. The probe resolution was coarse: 3.8 cm separation radially, and 3.7 cm separation axially. Vector $\mathbf{B}$ was measured at the 16 locations at a cadence of up to 65 MHz. The probe array was calibrated with a pulsed Helmholtz coil and tested with pulsed line currents. Data from the 2D probe array indicates whether magnetic reconnection on a given shot is likely or not. Unlike prior experiments [17], we had little control over the eventual orientation of the magnetic flux at the midplane in the present experiment.

We measured a line-averaged plasma density at the same axial location as the 2D array using a HeNe laser interferometer. The interferometer was calibrated with an initial phase shift of $\pi/2$ with the use of a Wollaston prism. Balance between the sine and cosine outputs of the Wollaston prism resulted in a circular pattern when one channel was plotted against the other. The SSX HeNe interferometer has become a very reliable diagnostic [6]. Our typical peak densities were $7 \times 10^{15}$ cm$^{-3}$ at the moment of merging, and the mean density was about $3 \times 10^{15}$ cm$^{-3}$ after merging.

Ion temperatures were measured both at the midplane and 0.24 m off the midplane using an ion Doppler spectroscopy (IDS) system with a 1 MHz cadence [18]. We measured emission from $C_{III}$ impurity ions and relied on the rapid equilibration of protons with the carbon ions ($\tau_{equil} \approx 20$ ns). Light from the $C_{III}$ 229.687 nm line collected from the plasma along a chord was dispersed to 25th order on an echelle grating and was recorded using a 16-channel photomultiplier tube (PMT). The collection beam was approximately 1 cm in diameter. The time-resolved proton temperature and line-of-sight average velocity were inferred from the observed thermal broadening and Doppler shift of the emission line, respectively. The line-of-sight was across the flow direction and was aligned on a radial chord. Our typical ion temperatures were $T_i \geq 20$ eV and up to 80 eV transiently.

Finally, we used vacuum ultraviolet (VUV) spectroscopy for line-averaged measurements of $T_e$ [19] in similar plasmas. The VUV spectroscopy was installed 0.05 m away from the gun (in the turbulence region) and was line integrated over a diameter. We found in those experiments that our electron temperature was about 7 eV for most of the discharge, and for most initial plasma gun conditions.

## 3. Experimental Results

### 3.1. Theoretical Background

One source of heating with the merger of two parcels of high-velocity plasma is the direct conversion of kinetic energy to heat:

$$\frac{1}{2}m_p v^2 = k_B T_p,$$

where we assume that the directed kinetic energy is thermalized into two degrees of freedom, ($T_\perp$ and $T_\parallel$). For example, two plasma plumes moving towards each other at 30 km/s (i.e., with a closing speed 60 km/s) would thermalize to $T_p = 20$ eV in a completely inelastic collision. This thermalization would happen very quickly. A 20 eV proton streaming in a $7 \times 10^{15}$ cm$^{-3}$ density plasma has a collision time of 30 ns [20], so it would thermalize in $\ll 1$ μs.

A second source of heating with rapidly merging magnetized plasmas is magnetic reconnection [21,22]. We found at SSX that reconnection proceeds at about 0.1 of the Alfvén speed [23], in other words, reconnection and the attendant heating takes about 10 radial Alfvén times ($\tau_A = R/V_A$). A similar reconnection rate has been observed in several other experiments [24,25] and simulations [26,27]. Indeed, a normalized reconnection rate of 0.1 appears to be ubiquitous in nature [28]. In our case, the Alfvén speed for $7 \times 10^{15}$ cm$^{-3}$ and $B = 0.3$ T is about 80 km/s, so a radial Alfvén time is about 0.08 m/80 km/s = 1 μs. We therefore expect reconnection heating to take at minimum 10 μs.

### 3.2. Counter- and Co-Helicity Merging

A typical shot is depicted in Figure 2. This is a counter-helicity merging shot displaying several key features. First, the two plasmas merge at the midplane, indicated by the large temporal peak in line-averaged midplane density of $8 \times 10^{15}$ cm$^{-3}$. This is followed after a significant heating delay by a peak proton temperature of $T_p = 75$ eV for this event. Ion temperature was measured 0.24 m off the midplane for this shot, but measurements at the midplane had similar behavior. Fluctuations of the magnetic field at the midplane are also shown.

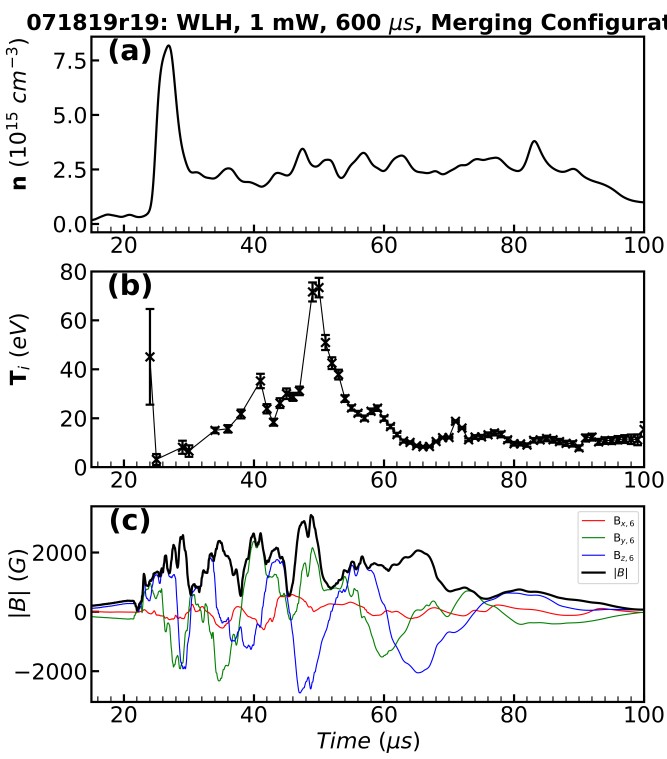

**Figure 2.** Counter helicity merging (single shot). Plot of (**a**) line-averaged density $n_e$, (**b**) proton temperature $T_i$, and (**c**) components and mean magnetic field on a typical probe, for counter-helicity merging. This event shows a peak merging density of $8 \times 10^{15}$ cm$^{-3}$ and a peak proton temperature of $T_p = 75$ eV.

Ensemble averages are depicted in Figures 3 and 4 for counter-helicity and co-helicity merging, respectively. These are 30 shot averages, temporally aligned at $t = 0$ (as opposed to, say, the density pulse). What we observed on average is similar to that shown in Figure 2. Both cases show a pronounced density pulse, followed by an increase in proton temperature. We found that the counter-helicity merging tends to have a wider spread in parameters, including some high-temperature pulses. Both cases display generally similar average behavior as summarized below.

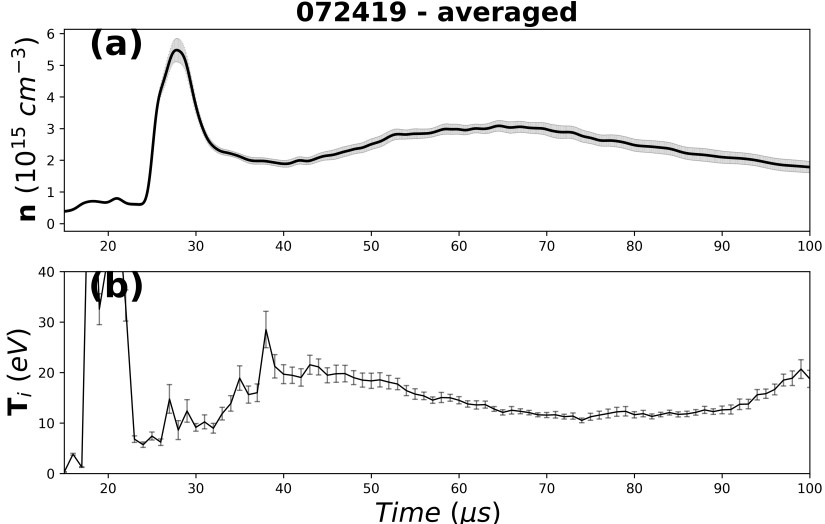

**Figure 3.** Counter helicity merging (average of 30 shots). Plot of (**a**) average density $n_e$, and (**b**) proton temperature $T_i$, for counter-helicity merging. Note that because of shot-to-shot jitter, the peak values plotted here tend to be lower than the statistics mentioned in the text.

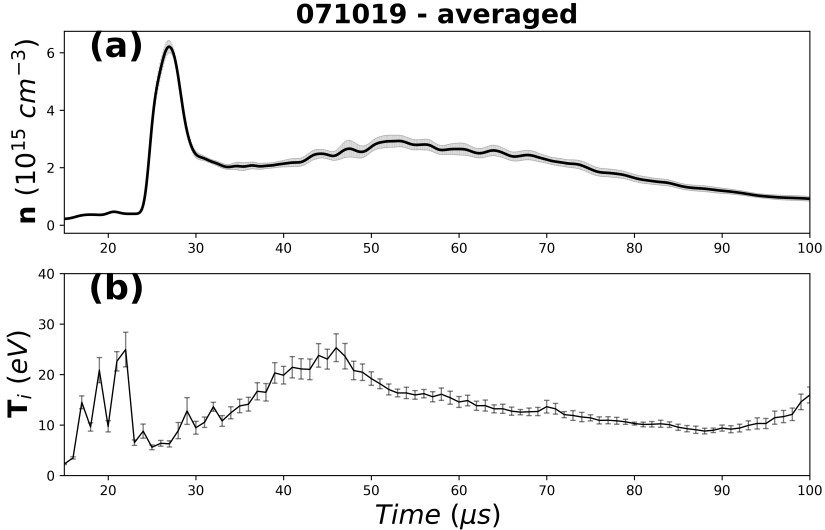

**Figure 4.** Co-helicity merging (average of 30 shots). Plot of (**a**) average density $n_e$, and (**b**) proton temperature $T_i$, for co-helicity merging. Note that because of shot-to-shot jitter, the peak values plotted here tend to be lower than the statistics mentioned in the text.

We found that for counter-helicity merging, the initial merging time (defined as the initial pile up of density for an ensemble of the 30 shots from Figure 3) is $27.9 \pm 0.2$ μs, with the mean peak density: $n_e = 6.4 \pm 0.3 \times 10^{15}$ cm$^{-3}$. The mean density after merging is $n_e = 2.2 \pm 0.1 \times 10^{15}$ cm$^{-3}$. We found that the mean heating pulse is $\Delta T_p = 24.0 \pm 2$ eV. The heating time, defined as the delay from initial merging to peak heating, is $13.9 \pm 0.7$ μs. The mean proton temperature after reconnection has terminated is $T_p = 15.6 \pm 0.6$ eV.

We found that for co-helicity merging, the initial merging time (defined as the initial pile up of density for an ensemble of the 30 shots from Figure 4) is $26.9 \pm 0.1$ μs, with the mean peak density $n_e = 6.9 \pm 0.2 \times 10^{15}$ cm$^{-3}$. The mean density after merging is $n_e = 2.9 \pm 0.1 \times 10^{15}$ cm$^{-3}$. We found that the mean heating pulse is $\Delta T_p = 20.1 \pm 0.3$ eV. The heating time for co-helicity merging is $17.7 \pm 0.7$ μs. The mean proton temperature after reconnection has terminated is $T_p = 15.1 \pm 0.5$ eV.

While the mean heating pulse in either case (about 20 eV) is consistent with thermalization of 30 km/s counter-flowing protons, the heating time is orders of magnitude longer. Indeed, the heating time is consistent with about 15 radial Alfvén times, or a normalized reconnection rate of $1/15 = 0.07$.

We studied 2D magnetic field movies for over 120 shots and found evidence of magnetic field reversal and reconnection in most of them. As discussed in our summary below, non-axisymmetric merging of Taylor states is significantly more complex than has been observed in typical axisymmetric reconnection experiments conducted at SSX [17,23,29]. Reconnection can occur anywhere on our $4 \times 4$ array, indeed reconnection could occur somewhere other than where we measure.

In Figure 5, we depict a magnetic reconnection event that occurs in the midplane. On this shot, a loop of magnetic flux swirls down and back to the left on the east side, and another flux loop is directed mostly radially and back to the right on the west side. This moment in the discharge (about 29 μs) is a few μs after merging, and marks the beginning of a protracted heating event of about 10 μs duration. This orientation persisted for a few μs before moving off the probe array. In Figure 6, we show the line-averaged data for the 2D map depicted in Figure 5. The dot indicates the moment of field reversal. It occurs a few μs after the two Taylor states merge. Heating proceeds for about 10 μs (i.e., $10 \, \tau_A$), reaching a peak temperature of 50 eV on this shot.

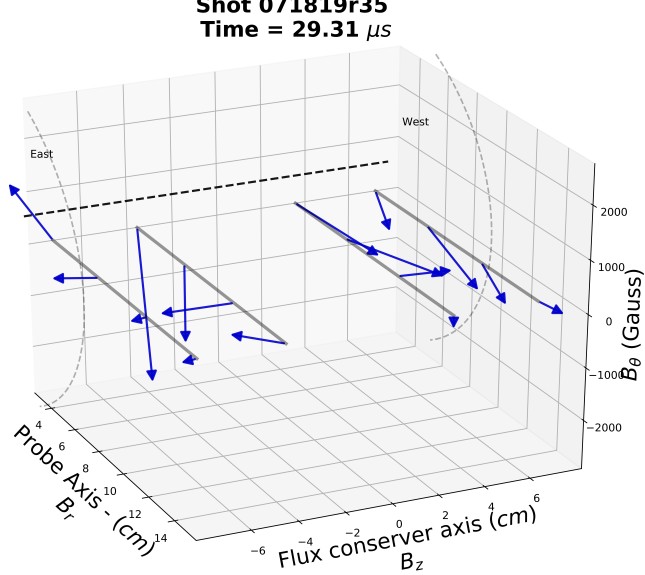

**Figure 5.** Field reversal from 2D probe array for a counter-helicity shot. Map of vector **B** at 16 locations at the midplane. This moment in the discharge (about 29 μs) is a few μs after merging, and marks the beginning of a protracted heating event of about 10 μs duration.

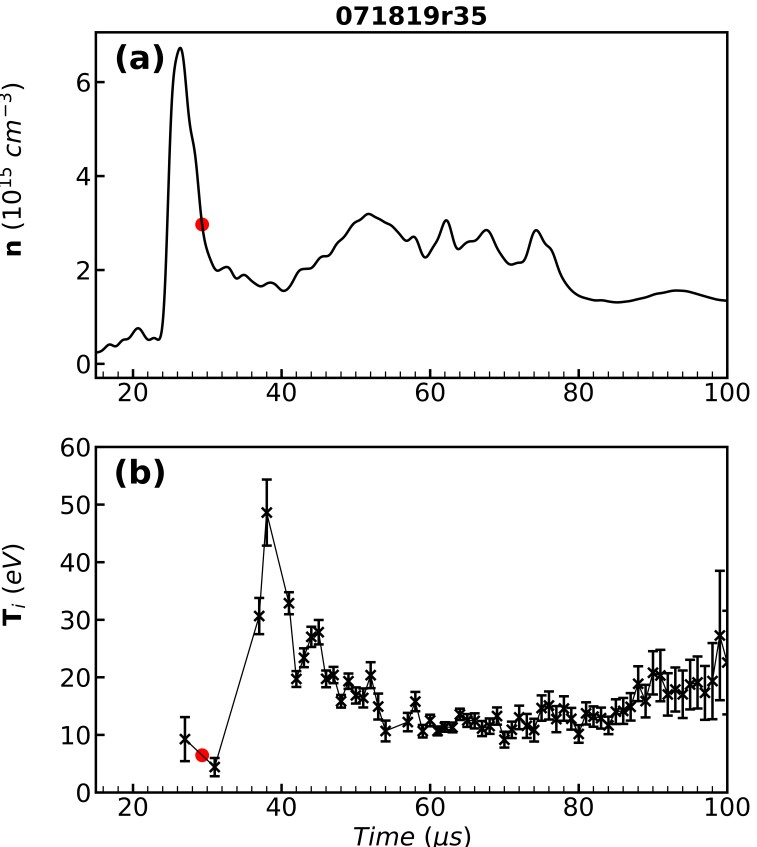

**Figure 6.** Line-averaged data for the 2D map depicted in Figure 5 . Plot of (**a**) line-averaged density $n_e$, and (**b**) proton temperature $T_i$. The moment of reconnection from Figure 5 is indicated by a dot. Note that this is just after merging of the east and west Taylor states, and marks the beginning of a 10 μs heating pulse, with a peak $T_i = 50$ eV.

## 4. Simulation Results

### 4.1. Dedalus Framework

Magnetohydrodynamic (MHD) simulations of these merging experiments were performed in the Dedalus framework. Dedalus solves differential equations with spectral methods written with a Python wrapper. It is an open source MPI. parallelized environment, http://dedalus-project.org/ [30]. The following normalized equations were advanced with initial spheromaks and high-density regions surrounding these spheromaks at both ends of a $2 \times 2 \times 10$ box.

$$\frac{\partial [\ln \rho]}{\partial t} + \nabla \cdot v = v \cdot \nabla [\ln \rho] \tag{1}$$

$$\frac{\partial v}{\partial t} + \nabla T - \nu \nabla \cdot \nabla v = T \nabla [\ln \rho] - v \cdot \nabla v + \frac{J \times B}{\rho} \tag{2}$$

$$\frac{\partial A}{\partial t} + \nabla \phi + \eta J = v \times B \tag{3}$$

$$\frac{\partial T}{\partial t} - (\gamma - 1)\kappa \nabla^2 T = -(\gamma - 1)T \nabla \cdot v - v \cdot \nabla T + (\gamma - 1)\eta |\mathbf{J}|^2, \tag{4}$$

where **B** is the magnetic field which is defined using the magnetic vector potential **A** ($\mathbf{B} = \nabla \times \mathbf{A}$) to enforce the Coulomb gauge ($\nabla \cdot \mathbf{B} = 0$), **J** is the current density and $\mathbf{J} = -\nabla^2 \mathbf{A}$ (here, $\nabla^2$ is the vector Laplacian), $\rho$ is the number density of the plasma, T is the temperature of the plasma, $\gamma$ is the adiabatic index for monoatomic ideal gas and $\gamma = \frac{5}{3}$, $\eta$ is resistivity, $\nu$ is kinematic viscosity, $\kappa$ is heat conductivity, and $v$ is the velocity of the plasma. The pressure of the plasma is given by $P = \rho$ T.

The simulations were performed with resistive MHD effects; Hall effects were ignored. The equations were normalized with the following constants. The unit length $R_0 = 0.08$ m, the number density $n_0 = 10^{16}$ cm$^{-3}$, and the unit magnetic field $B_0 = 0.5$ T. The Alfvén speed was 10.9 cm/μs and this means the unit time was $R_0/V_A = 0.73$ μs. The resistivity, viscosity, and heat conductivity were kept constant across the channel as a simplifying assumption, where $\eta = 0.001$, $\nu = 0.05$, and $\kappa = 0.01$, respectively. In practice, the resistivity is expected to be higher at the walls than the core of the channel due to the high temperature at the core of the channel.

The simulations were defined over a rectangular domain $(x, y, z) \in [\pm 1, 0, 0] \times [0, \pm 1, 0] \times [0, 0, 10]$. The simulations were preformed over rectangular mesh $(n_x, n_y, n_z) = (28, 24, 180)$ with parity (sin/cos) basis in all three directions. The walls were assumed to be perfectly conducting enforced by the condition $\nabla \times \mathbf{A} = 0$, and we had free-slip boundary conditions at the walls. The simulations were initialized with spheromaks at each end. The structure of the magnetic field of the spheromaks was initialized by numerically solving the equation $\nabla^2(A_0(x, y, z)) = -J_0(x, y, z)$ in Dedalus. The simulations were performed for both co- and counter-helicity spheromaks and the final magnetic structures of these merging simulations are shown below.

### 4.2. Merged Taylor States

We present here the results of low resolution Dedalus simulations of Taylor state merging. The colored lines represent magnetic field lines. In Figure 7, we can see three frames from a counter-helicity merging simulation, depicting two Taylor states about to merge, then merging, and finally relaxed to a new state. Note that the final counter-helicity state has little twist. In Figure 8, we see three frames from a co-helicity merging simulation, depicting two Taylor states about to merge (5 $\tau_A$), then merging, and finally relaxed to a new state. Note that the final co-helicity state retains its twist, and that merging is not yet complete at 20 $\tau_A$. We will continue to study Taylor state merging in the Dedalus framework at higher resolutions and for longer durations. These efforts will be reported in a separate study focusing on the simulation details.

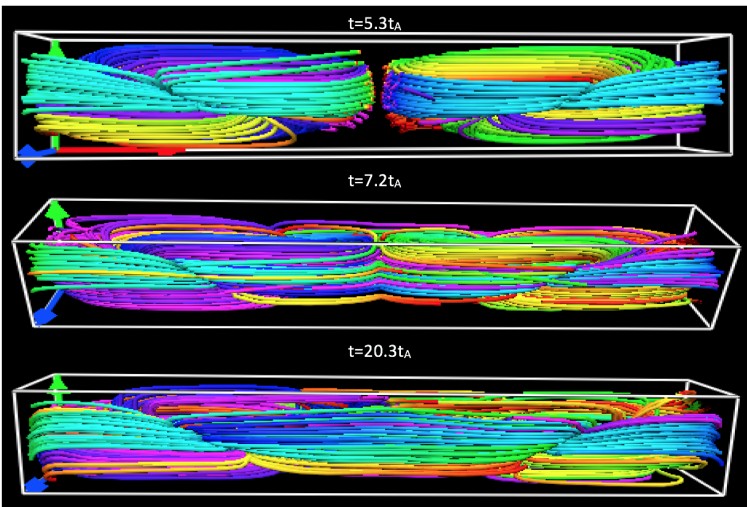

**Figure 7.** Low resolution simulation of counter-helicity merging using the Dedalus framework. The first frame shows relaxed Taylor states about to merge at about 5 $\tau_A$. The second frame depicts merging. The third frame depicts a relaxed merged state at about 20 $\tau_A$. The colored lines represent magnetic field lines.

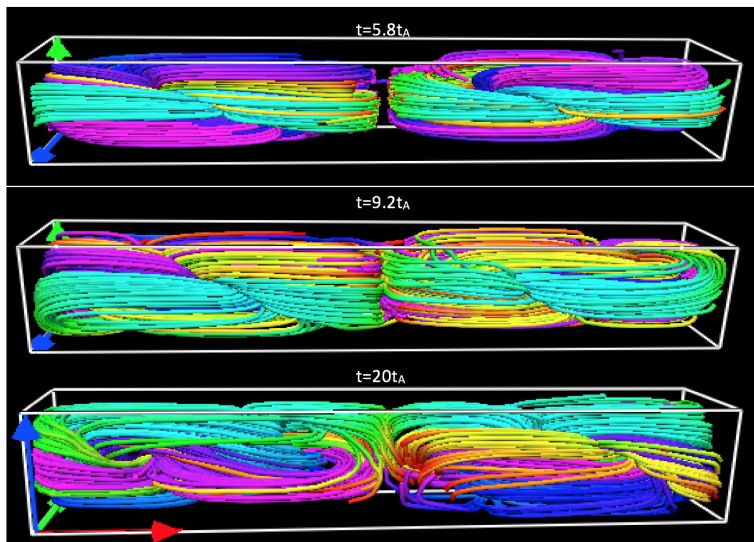

**Figure 8.** Low resolution simulation of co-helicity merging using the Dedalus framework. The first frame shows relaxed Taylor states about to merge at about 5 $\tau_A$. The second frame depicts merging. The third frame depicts a relaxed merged state at about 20 $\tau_A$. The colored lines represent magnetic field lines. We believe this relaxed state is likely to have better confinement properties than the counter-helicity state in Figure 7 .

## 5. Discussion and Summary

Merging of Taylor states is considerably more complex than earlier experiments performed at SSX with axisymmetric spheromaks [17,29]. First, a Taylor state is non-axisymmetric so either merging object can have any rotational orientation (see Figure 1). Second, since some distance between the source and target region is required for the Taylor state to form and relax (typically five flux conserver radii), and since there is some variation in the flow speed ($\pm 5$ km/s typically), the merging location can vary $\pm 5$ cm off the midplane. Indeed, we see evidence of the reconnection layer forming at various locations around our 2D probe array. In addition, since the heating time is about 15 μs, a reconnection site typically forms, then moves off the probe array before peak heating is recorded. Third, since the plasmas do not have a flux conserving boundary at the midplane, it is possible for the Taylor states to slide off-axis and merge in complex ways. Finally, we expect the reconnection layer to be only a few $\delta_i = c/\omega_{pi} = 4$ mm thick. We cannot resolve this with our probe array, so our indication of a reconnection event is a field reversal between some set of probes, along with ion heating and density pile up. Since our ion Doppler spectrometer collection beam is about 1 cm wide, capturing reconnection outflows and local heating is difficult in this configuration [29].

To summarize our findings, we define the instant of merging as the time of density pile up at the midplane (typically 27 μs after the plasma guns are fired). Since the Taylor states are ejected about 14 μs after the guns fire, the time of flight is about 13 μs for 0.5 m to the midplane. This gives us a typical velocity of 38 km/s or 3.8 cm/μs. We observed several reconnection scenarios but a clear example at the midplane is depicted in Figure 5. We observe what appears to be reconnection-driven ion heating of about $\Delta T_p = 20$ eV that takes place in about 15 μs. Some events generate transient temperatures up to 80 eV.

While the counter-helicity shots show somewhat more dynamical heating, the final co-helicity merged state has better confinement properties than the counter-helicity state. Our assessment is that if a merged, stagnated, hot Taylor state should be implemented as a target for future magneto-inertial fusion experiments, co-helicity merging will generate a new Taylor state with the same twist as the initial objects. A Taylor state with net helicity should have better confinement properties than a zero

helicity state. In any case, both scenarios generate similar densities and temperatures on average (Figures 3 and 4).

Low resolution simulations using the Dedalus framework reveal the dynamics and final merged state. Since the simulation is low resolution (corresponding to $\geq$ 5 mm), and single-fluid MHD, we do not expect to capture the dynamics of the reconnection layer. More sophisticated simulations are planned, but on larger scales, the Dedalus simulation provides guidance as to the structure of the final relaxed state.

**Author Contributions:** The lead author was in charge of conceptualization, supervision, original draft preparation, and editing, M.B. Formal data analysis, K.G. Simulation and software, M.M. All authors have read and agreed to the published version of the manuscript.

**Funding:** This research was funded by DOE Advanced Projects Research Agency (ARPA) ALPHA program project DE-AR0000564. Computations were performed under the NSF XSEDE research allocation PHY190003 at the Bridges Supercomputer.

**Acknowledgments:** This work was supported by the DOE Advanced Projects Research Agency (ARPA) ALPHA program project DE-AR0000564. Computations were performed under the NSF XSEDE research allocation PHY190003 at the Bridges Supercomputer. The authors wish to acknowledge the support and encouragement of ARPA program managers Patrick McGrath and Scott Hsu. We would like to particularly acknowledge undergraduate student contributions from Lucas Dyke, Nick Anderson, Hari Srinivasulu, Emma Suen-Lewis, Luke Barbano, and Jaron Shrock, and technical discussions with colleagues Jeff Oishi, David Schaffner, and Adam Light. Special thanks to Manjit Kaur for constructing the 2D probe array. Technical support from Steve Palmer and Paul Jacobs at Swarthmore for SSX is also gratefully acknowledged.

**Conflicts of Interest:** The authors declare no conflict of interest. The funders had no role in the design of the study; in the collection, analyses, or interpretation of data; in the writing of the manuscript, or in the decision to publish the results.

## Abbreviations

The following abbreviations are used in this manuscript:

| | |
|---|---|
| MHD | magnetohydrodynamics |
| MIF | magnetoinertial fusion |
| SSX | Swarthmore Spheromak Experiment |
| IDS | Ion Doppler spectroscopy |
| VUV | vacuum ultraviolet. EOS equation of state |
| PMT | photomultiplier tube |
| HeNe | Helium-Neon |

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
