# Peer review of "Taylor State Merging at SSX: Experiment and Simulation"

_plasma, doi:10.3390/plasma3010004_

Round 1

Reviewer 1 Report

The manuscript presents a fine review of the work of SSX, relative to the experiment and the simulation studies relative to it. I do not see the need for improvements of the manuscript that can be published in its present form.

Author Response

Thank you for your approval.  Nonetheless, we strive to improve the manuscript further by addressing the concerns of the other referees.

Reviewer 2 Report

Comments on manuscript “Taylor State Merging at SSX: Experiment and Simulation” by M R Brown et al
The paper presents some new and original results which will be useful to publish. However, some
work should be done on the readability of the paper.
1. The title, "Taylor state merging..." is confusing. The paper describes merging of two plasma
rings, not "states". I recommend to change the title and replace "state" to some object (i.e.
plasma rings, spheromaks, Taylor state plasmas etc) in the text. Of course, the meaning of
the Taylor state is well known and probably describes the plasma in SSX.
2. Introduction. I am not sure about relevance of MIF to the presented research. However,
merging experiments i.e. ref 24 are much more relevant.
3. P2 line 57. “we have opened a 0.18 m gap” => please explain. Have you broken the vacuum?
Could you show in Fig 1?
4. P.2 line 67. Large flow speeds => could you specify the direction of rotation if plasma is nonaxisymmetric?
5. P.3 line 92. It may be useful to show the position and the view in Fig.1? Also, what means “in
similar plasmas”? How similar, what was the difference, if relevant?
6. P.3 line 105. “radial Alfven times” => do you mean “poloidal”, or “toroidal”?
7. P.3 line 106. I am not sure that in ref 24 reconnection proceeds at about 0.1 of the Alfvén
speed, typical timescale was tens of microseconds. Please check. Probably better to say
“similar reconnection rates”.
8. P.3 lines 112, 114. Could you show in Fig.1 where Ti was measured and where is the density
peak? Do you mean peak in time or in space?
9. P.5 line 192. Could you show in Fig.1 location of 2D probe array? Only 1D location is shown
unless lines correspond to 2D location.
10. P.5 line 193. “reconnection site typically forms, then moves…” What do you mean by “site”?
11. P.5 line 195. Plasmas, or plasma filaments can merge, but not “states”. See previous
comments. What do you mean by “complex ways”? Please explain.
12. P.5 line 205. “some events” – please specify which events.
13. P.6 line 211. “same twist of the
14. initial objects” – Do you suggest that helicity is fully conserved during merging process, or
some of it is lost?
15. Fig.3,4. Please explain the peak in Ti that happens before the reconnection.
16. Please explain what exactly is shown in Fifs.7,8. Flux lines, field lines? What does the color
mean?

please see attached.

Author Response

{\bf Referee 2:} The referee seems to have missed a key point that we will clarify in the revised manuscript. It is true that years ago, we merged plasma rings (called spheromaks) in the SSX device. We use the term ``Taylor state'' to refer to a different structure, one that has relaxed and twisted in a non-axisymmetric way. Throughout the referee refers to this older work (eg confusing toroidal/poloidal with radial). We will address the concerns here.

{\bf 1, 2, 6:} We have added the clarification to the introduction: Taylor states in SSX begin as axisymmetric spheromaks (with poloidal and toroidal fields) but quickly evolve via instabilities to a twisted, non-axisymmetric structure bounded by a cylindrical copper liner inside the vacuum chamber. The key benefit of Taylor states is that they are stable; they have already undergone the instabilities that plague other configurations.

{\bf 3, 9:} The gap and the location of the four probes forming the 2D array is clearly shown in Figure 1. Of course, we don't break vacuum.

{\bf 4:} The flow speed is directed away from the source as in a wind tunnel. Again, no poloidal or toroidal here.

{\bf 5:} We have included a reference if the reader wishes to learn more about the VUV spectrometer (not a key point here).

{\bf 7:} We have changed to ``Similar reconnection rates...''

{\bf 8:} We have added: ``but measurements at the midplane had similar behavior.'' The point is that all measurements are nearly at the same location. We have added ``temporal peak''.

{\bf 10-14:} The referee points out that the merging of these twisted flux-rope Taylor state structures is complex. We understand that and have presented what we observe. We state that ``our indication of a reconnection event is a field reversal between some set of probes''. This is what we mean by a site. We have listed what we mean by complex. Reconnection angle, axial location, etc can all vary from shot-to-shot. The sentence reads: the same twist of the initial objects. We have not measured helicity but the dynamics suggest that helicity must be better conserved than energy. Helicity sure decays slowly during the experiment.

{\bf 15:} We don't understand the broad impurity line before merging.

{\bf 16:} We have added the phrase: The colored lines represent magnetic field lines.

Reviewer 3 Report

The article presented measurements of plasma density, temperature and magnetic field of the Taylor state merging at SSX. It also preformed Dedalus simulation for both co- and counter-helicity merging. While I believe the experimental data and simulation of the merging have value to the audience, the authors may want to consider my comments to improve the clarity of the paper and also the impact.

It is not very clear to me what are the major points that authors want to make. Is the theme of this paper a validation of plasma heating mechanisms? Or is it to identify which heating mechanism is dominant? Or is it an experiment-simulation comparison? The authors have written plenty of measured values in the paper, for example, the first two paragraphs of page 4. But it is less clear to me what is the meaning of the data and the physics implication of them. The authors presented the experiment and simulation as two separated pieces of stories. Although I understand that precise comparison is very difficult, they may want to comment on something qualitative or at least the link between them.

Author Response

We thank the referee for their comment. The manuscript is not meant to be a simulation/experiment comparison. This preliminary simulation is meant as guidance as to what might be happening globally in our experiment. Our main point is to demonstrate the prospect of merging these interesting structures to form a hot target for magneto-inertial fusion. We have clarified this in the abstract and introduction.

Abstract: ``We suggest that the merged, warm state could be suitable target for future magneto-inertial fusion experiments.''

Introduction: ``... we have studied the dynamical merging of two Taylor state plasmas, seeking to form a hot, stable, stagnant configuration that could be a suitable MIF target. A stable, warm magnetized target plasma is needed for future MIF fusion experiments.''

Round 2

Reviewer 3 Report

The authors have addressed my concerns. I agree the work is now suitable for publication.